# Enhancing Syslog Message Security and Reliability over Unidirectional Fiber Optics

**DOI:** 10.3390/s24206537

**Published:** 2024-10-10

**Authors:** Alin-Adrian Anton, Petra Csereoka, Eugenia Ana Capota, Răzvan-Dorel Cioargă

**Affiliations:** Computer and Information Technology Department, Faculty of Automation and Computing, Politehnica University Timișoara, 2nd Vasile Pârvan Ave., 300223 Timișoara, Romania; petra.csereoka@upt.ro (P.C.); eugenia.capota@upt.ro (E.A.C.); razvan.cioarga@upt.ro (R.-D.C.)

**Keywords:** syslog, air gap, log management, log monitoring, data diode

## Abstract

Standard log transmission protocols do not offer a robust way of segregating the log network from potential threats. A secure log transmission system and the realization of a data diode using affordable components are proposed. Unidirectional data flow prevents unauthorized access and eavesdropping, ensuring the integrity and confidentiality of sensitive log data. The system uses an encryption protocol that requires that the upstream and the downstream of the data diode are perfectly synchronized, mitigating replay attacks. It has been shown that message amplification can mitigate UDP packet loss, but this is only required when the data diode traffic is congested. The implementation of the encryption algorithm is suitable for resource-constrained devices and it has been shown to produce random-looking output even on a reduced number of rounds when compared to the parent cipher. Several improvements have been made to the original encryption algorithm for which an actual implementation was missing. Free software and datasets have been made available to reproduce the results. The complete solution is easy to reproduce in order to secure the segregation of a log network inside any scenario where logging is required by the law and log tampering must be prevented.

## 1. Introduction

In the modern digital landscape, ensuring the secure and reliable transmission of log data is paramount to maintaining the integrity and availability of information systems. System logs, commonly referred to as syslog messages, play a critical role in monitoring, diagnosing, and troubleshooting network and system activities. However, the transmission of these messages over standard network protocols, such as User Datagram Protocol (UDP), presents significant challenges due to their inherent lack of reliability and security. This paper addresses these challenges by proposing a novel encrypted syslog redirector and amplifier designed to enhance the security and reliability of syslog message transmission through the use of a data diode.

The primary motivation behind this research comes from the need to protect sensitive log data from possible intercept and loss. In environments where system logs are transmitted over potentially insecure networks, the risk of eavesdropping and data loss can lead to severe security breaches and operational disruptions. Traditional log transmission methods often fail to provide the necessary level of security and reliability. UDP, despite its efficiency, does not guarantee message delivery, which can result in critical log messages being lost in transit. Furthermore, the lack of encryption in standard syslog transmission makes the messages vulnerable to interception and unauthorized access.

Most Internet of Things (IoT) devices do not have the computing power to analyze logs, and even so, a centralized approach offers a broader understanding. The Australian Security Directorate and international partners provide guidelines for best-practice logging [1,2,3].

Given the potential for a huge influx of data from logs, having a strong log management strategy is crucial. Implementing security information and event management (SIEM) systems like Wazuh [4] along with security orchestration, automation, and response (SOAR) tools like The Hive [5] to effectively manage and process these data into actionable alerts is of critical importance [6].

### 1.1. Related Work

Network separation was identified as the most efficient solution for protecting high-security networks [7]. For resource-constrained networks such as IoT, lightweight cryptography was applied to protect the confidentiality and authenticity of the data [8,9]. Huffman encoding and AES encryption were combined for data transmission optimization [10]. Block ciphers were used in counter mode to mitigate packet spoofing and packet replay [11]. For the end-to-end encryption of medical IoT and patient data, homomorphic encryption (HE) is a paradigm shift [12].

Several centralized log management solutions have been built upon the foundation of syslog, offering robust log management capabilities. Loggly delivers a cloud-based centralized solution specifically designed to present a single source of truth [13]. Graylog provides a free community edition for flexible log aggregation and analysis [14]. Logging Made Easy (LME), developed by the Cybersecurity and Infrastructure Security Agency (CISA), is a user-friendly, no-cost log management tool designed for small to midsize organizations with limited resources [15]. It provides centralized monitoring, real-time insights, and threat detection. Additionally, software tools such as Clickhouse 24.10.1.1245, Loki 3.1.1, and Logstash 8.15.2 offer centralized syslog management with advanced search and monitoring capabilities [16,17,18].

For very large log databases, μSlope offers both compression and search capabilities on partially uncompressed data [19]. Searchable encryption has been applied to audit logs to prevent logging manipulation and data exfiltration [20].

The authors of [21] present various log-as-a-service solutions that aim to improve log management and log data retention. Figure 1 is based on their view on a generalized cloud log forensics system. As mentioned by them, there are several laws pertaining to logging requirements, spanning a range from NIST standards like SP800-53 and SP800-66 to PCI DSS and Gramm–Leach–Bliley Act (GLBA), which requires financial institutions to maintain proper log management.

The provenance of log data is of great importance and is more visible when dealing with the provenance of log data in a cloud [22]. The authenticity and validity of the information stored in a logging system may be as important as being able to debug and fix observable failures and also to mitigate an ongoing attack.

### 1.2. Standards

The syslog protocol serves to transport event notification messages across IP networks to syslog servers. It allows machines to send messages without specifying the content or format of the message, allowing for flexibility in message handling. The protocol is designed to facilitate the collection and distribution of event messages between networked devices [23]. This protocol lacks mechanisms for reliable message delivery, which could lead to lost or intercepted messages. Messages can be damaged, replayed, or modified in transit, compromising their integrity. In addition, there are concerns about message observation, prioritization, and differentiation, as well as the risk of misconfiguration and forwarding loops, posing security risks for network operators.

RFC 5424 recommends the deployment of Transport Layer Security (TLS) support for enhanced security [24]. It allows for the use of additional transport protocols and offers a foundation on which to build syslog extensions, addressing compatibility issues. The implementation of TLS is described in RFC5425 [25], and the transport of UDP is specified in RFC 5426 [26]. A syslog application may know when messages were dropped by keeping track of the messages it receives and the ones it successfully transmits. If there is a discrepancy between the received and transmitted messages, the application can infer that some messages were dropped. Additionally, the syslog application can implement mechanisms to monitor and detect message loss, triggering notifications to inform collectors or relays about lost messages [24].

RFC 3195 [27] and RFC 6587 [28] establish a standard for the transmission of syslog messages over the transmission control protocol (TCP) [29,30] to mitigate the reliability problem using TCP acknowledgments and retransmission mechanisms [31].

The main difference between the old Berkeley Software Distribution (BSD) syslog (RFC3164) and the new format is the quality of the timestamp, the format of the message, and the maximum size of a message.

RFC6012 [32] introduces Datagram TLS (DTLS) [33] for syslog messages, which has been mapped on UDP [34] and on the Datagram Congestion Control Protocol (DCCP) [35].

The flow control mechanisms present in protocols like TLS, DTLS, and DCCP require bidirectional communications between the syslog sender and the receiver.

RFC5848 [36] introduces a way to digitally sign syslog messages to provide the authenticity of the message.

To cope with the provenance problem, a public-key cryptography-based system is proposed in [37]. The authors define several security requirements for log data in transit and for log data at rest.

The requirements for log data in transit according to [37] involve origin authentication, which means that the log concentrator must ensure that log packets were sent by authorized devices; also, the message must be confidential, must preserve integrity, must be unique, and should be delivered in a reliable way.

For the integrity of the log data at rest, entry accountability and confidentiality are claimed to be sufficient requirements [37].

The DTLS protocol strengthens security in client–server communication by preventing eavesdropping, tampering, and message forgery. It provides security assurances comparable to TLS but is tailored for applications using UDP rather than TCP. DTLS ensures secure communication by maintaining the semantics of the datagram and minimizing the need for new security mechanisms, making it an ideal choice for safeguarding data on the Internet [38].

The choice of UDP delivery over TCP for syslog messages is related to the historical simplicity of the datagram protocol and, consequently, to the ease of packet generation and forwarding for small text messages, especially by devices with constrained hardware resources. On the other hand, the problem of dropped packets is not always as critical as it may seem because failures and other types of warnings are going to trigger several messages to be sent.

In modern times, the reliability of message delivery and the confidentiality and authenticity of the data in transit are of increasing importance.

A log management infrastructure typically comprises the following three tiers [2]:Log generation;Log analysis and storage;Log monitoring.

To address the issues of packet loss and confidentiality, we introduce a syslog transmission framework that leverages encryption and redundancy over a unidirectional fiber optic link, commonly known as a data diode. The simplest arrangement for the second tier is a single log server that handles all log analysis and storage functions [2]. This data diode ensures a one-way data flow, thereby preventing any reverse communication and enhancing security.

The rest of this paper is organized as follows: the Methods section details the design and implementation of the syslog redirector and amplifier, including the encryption mechanisms, deduplication process, and configuration of the data diode; the Results section presents the performance evaluation of our system, demonstrating its efficacy in terms of reliability, security, and throughput; the Discussion section explores the implications of our findings, potential limitations, and avenues for future research; the Conclusions section summarizes the main contributions of this work.

## 2. Methodology

### 2.1. Speck-R Enhanced

In order to secure communication over the unidirectional fiber optic link, the Speck-R [39] ultra-lightweight cipher summarized in Algorithm 1 has been modified as shown in Algorithm 2.

The RC4 vulnerable key scheduler [40] in Algorithm 3 has been replaced with the secure RC4D [41] version from Algorithm 4. RC4 weak keys and other vulnerabilities have been studied for a long time, making RC4 a suboptimal choice for use in Speck-R [42,43,44,45].

The enhanced Speck-R substitution boxes generated using RC4D [41] are a password-depending secret. Line 8 in Algorithm 2 shows that a hash of the password is used in order to initialize the algorithm. Ciphers with known substitution boxes as Speck-R [39], or those that do not require substitution boxes at all, like Simeck-Tea [46] or SIMON and SPECK [47], are lighter on resources but may not provide the same level of security.

The main idea behind Speck-R [39] is that a reduced number of ordinary SPECK [47] rounds shown in lines 10–12 are followed by a dynamic substitution boxes routine in line 18, where 3 susbtitution boxes are interchanged depending on the block count. These boxes are initially configured with the RC4 Algorithm 3. Another particularity of Speck-R is the swap used in the process of counter incrementation—line 13 in Algorithms 1 and 2. Swapping two 32-bit words from the 64-bit counter followed by an increment in the right part makes sure that the ciphertext is more scrambled.

The LOOP variable introduced in Algorithm 2 ensures that the full 26 words of the expanded rounds key are used when encrypting multiple 64-bit blocks using an encryption key size of 96 bits.

Figure 2 emphasizes another modification in the enhanced version of Speck-R, where a round key of 832 bits is used in Figure 2b instead of a truncated 56-bit rounds key in Figure 2a.
**Algorithm 1:** The encryption process of Speck-R from [39] with vulnerable KSA_Init [42,43,44,45].
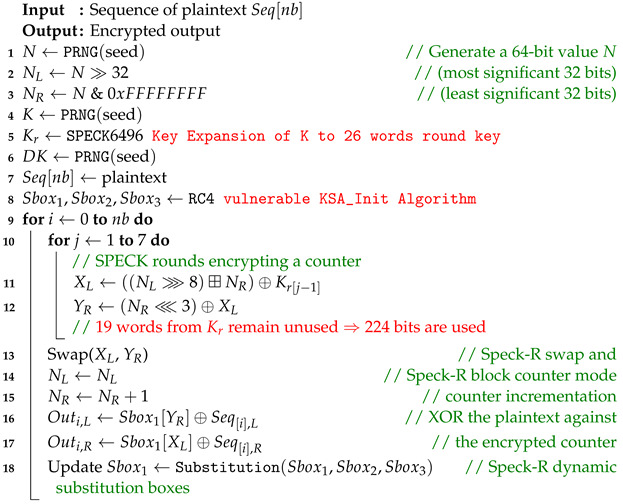


**Algorithm 2:** Our Enhanced Encryption process of Speck-R implemented in [48] with secure KSA_Init [41].

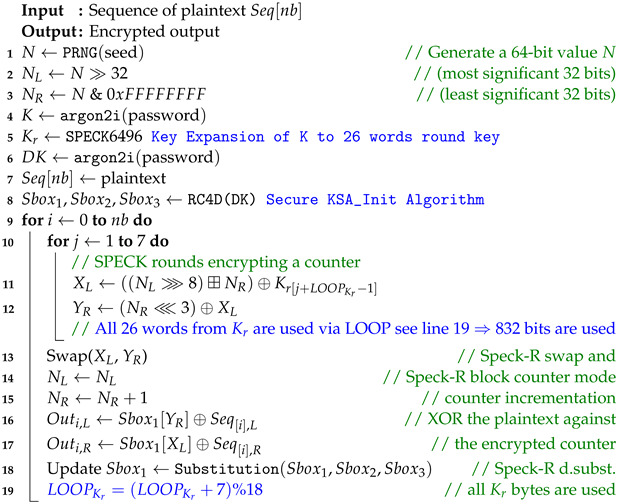



**Algorithm 3:** RC4 Key Scheduling Algorithm [40] used by Speck-R [39].

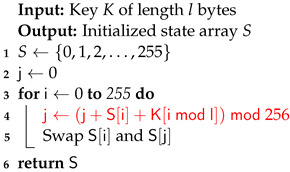



**Algorithm 4:** RC4D Key Scheduling Algorithm [41] used by Enhanced Speck-R.

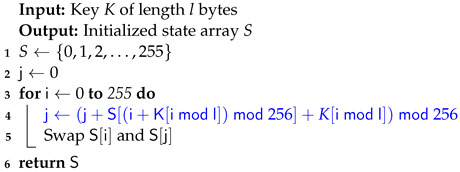



### 2.2. Data Diode Configuration

A data diode effectively drops any packet flowing in the wrong direction. A hardware data diode like those made of fiber optics simply does not present a backward communication channel. The GNU/Linux operating system is used with static arp [49] configured correctly on both sides of the data diode.

The medium access control address (MAC) is a unique identifier assigned to a network interface controller [50]. On Debian-based distributions, the arp tool is located in the net-tools package, which must be installed:


apt update



apt install net-tools


On host A, the IP address and the MAC address of the side B network card are added in the /etc/ethers file:


/etc/ethers (on side A):



bb:bb:bb:bb:bb:bb 192.168.10.2


On host B, the IP address and the MAC of the side A network card are added in the /etc/ethers file:


/etc/ethers (on side B):



aa:aa:aa:aa:aa:aa 192.168.10.1


On a bidirectional connection, these settings can be communicated freely between hosts, but on a unidirectional link, the MAC address information must be configured by the user.

The two network cards must be configured with IP addresses that are part of the same network.

For host A:


sudo ip addr add 192.168.10.1/24 dev enp1s0


For host B:


sudo ip addr add 192.168.10.2/24 dev enp1s0


If any network manager is running in the background, it can be disabled to avoid configuration interference; otherwise, the static IP address and the network mask must match.

The /etc/ethers file has to be loaded at boot time using the “arp -f /etc/ethers” command. Older distributions that use SysVinit will start all system services defined in scripts under /etc/init.d/, and then will end up running /etc/rc.local. This file may be edited to run the following:


/etc/rc.local:



/usr/sbin/arp -f /etc/ethers


and permissions should allow for the execution of


chmod +x /etc/rc.local


In order to create rc.local compatibility with systemd distributions, a systemd service unit must be created:


sudo nano /etc/systemd/system/rc-local.service


containing the following lines:


[Unit]



Description=Run script after network (on boot)



ConditionPathExists=/etc/rc.local



After=network.target



[Service]



Type=forking



ExecStart=/etc/rc.local



TimeoutSec=0



StandardOutput=tty



RemainAfterExit=yes



[Install]



WantedBy=multi-user.target


The service will start at boot but it has to be enabled:


sudo systemctl enable rc-local


and it can also be started immediately using the following command:


sudo systemctl start rc-local


Figure 3 describes the connection schematics of a data diode made of gigabit ethernet media converters. Each of the media converters also shown in Figure 4 has separate ports for transmission (TX) and reception (RX).

Fiber optic cables are connected from (TX) to (RX), but one of the media converters is placed between the other two (on top in Figure 4), making sure that the port (RX) from the input of the data diode will detect an on-line signal but will always remain disconnected from the output of the data diode.

Figure 5 shows the schematics of a fast ethernet data diode made of 2 TP-Link MC100CM media converters (Shenzen, China).

A fiber optic splitter is also visible in Figure 6 as a white cable with 2 sockets on one side and 1 socket on the other. It was used to copy the light signal back to the transmitter media converter.

This approach did not work for TP-Link MC200CM (Kowloon, Hong Kong) gigabit ethernet media converters shown in Figure 3 and Figure 4. In this case, the third media converter was used to emulate the light signal for the transmitter. The connection between the output media converter and the signal emulator is necessary in order to keep the signal emulator running.

The quality of the cable splitter influences the strength of the light signal over the fiber optic return path link. If the signal is too low at the (RX) port, the sender media converter may not detect an online connection. In addition, some protection schemes might simply prevent such configurations on more recent products.

### 2.3. Proposed Solution

The proposed framework consists of two main components: a syslog collector–sender, which encrypts and transmits syslog messages multiple times to ensure delivery, and a syslog receiver–forwarder, which deduplicates, decrypts, and forwards the messages to a rsyslog server [51]. The proposed solution is shown in Figure 7.

Several applications and devices are sending syslog messages to a syslog server residing on a network. The syslog server encrypts and boosts these messages by sending them in a repeated manner over a UDP protocol through a data diode. The data diode connects the logging network with a higher-security network in a unidirectional way. Incoming messages are filtered in a log aggregator to eliminate duplicates and decrypt them back to their original form. This high-security network allows for the live tailing and further monitoring of the information provided from outside the security level. This is basically a classic Bell–LaPadula security model (BLM), where the low network cannot read from the high network [52]. The storage devices in the high network can be of the write-once read-many (WORM) types such that log tampering is impossible even from the high network by mistake.

To summarize the Bell–LaPadula confidentiality concept, which also involves a BIBA integrity model (BIM) in Figure 7, no user in the low-security level network may read or modify information stored in the high-security level network [52]. This, of course, has to do with logs at rest. For data in transit, bogus devices can still send any kind of information that they want and can even try to replay logs sent by other devices. Replaying can be detected by the decryption process because the resulting text line will not contain only printable characters from the ASCII table but the authenticity and validity of the logged information will remain unknown and highly depend on the security of the devices and systems in the data center network themselves.

The message format for the logs that will be encrypted and boosted on the data diode is shown in Figure 8. A 64-bit counter is inserted as a prefix before encryption. This counter is to be removed prior to decryption. Inserting a fake counter by bogus devices is not possible since only the first 64 bits are going to be interpreted as a counter and this counter is always inserted by the sender. This is why a replayed message will have a bogus counter and decryption will result in garbage.

Algorithm 5 uses a repeatfactor constant in order to boost the sending of each message by repeating it a number of times (i.e., 100–1000). A counter is prefixed in front of each message such that the decoder can eliminate duplicates, automatically making sure that each message packet is going to be processed exactly once at the receiver side. This is of utmost importance for the decryption algorithm since internal counters in the cryptography routines in the default counter mode of operation (CTR) greatly depend on the fact that the inserted packet counters are unique.

Algorithm 6 describes the receiver side, where, for the sole purpose of counting missing packets, the decrypted lines are stored in a text file. The receiver software version 1.00 available on Github [53] relays the original syslog messages to a local syslog instance.

The script in Algorithm 7 is used to detect the number of missing packets, when syslog messages are sent through the data diode.

In the source code available on Github [53], the message counter works as shown by Algorithm 8.

For encryption, a context variable is used to hash the encryption password using Argon2i [54] and keep track of all counters. This is shown in Algorithm 9. This version of the Argon2 password hashing competition winner (PHC) [55] is memory-independent, making a brute-force attack much more difficult to carry out. The encryption password has been replaced with the serial number of the data diode in [53].
**Algorithm 5:** UDP Sender Algorithm.
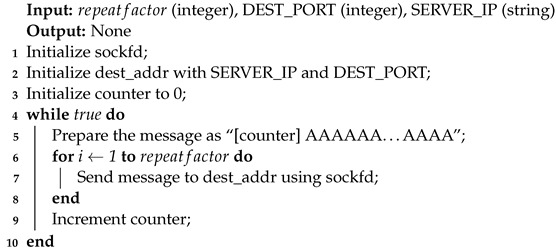


**Algorithm 6:** UDP Receiver Algorithm.

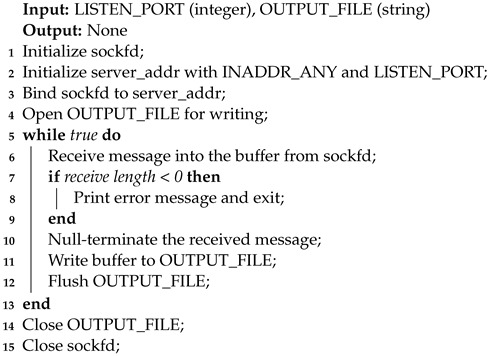



**Algorithm 7:** Message Counter Script Algorithm for Measurements.

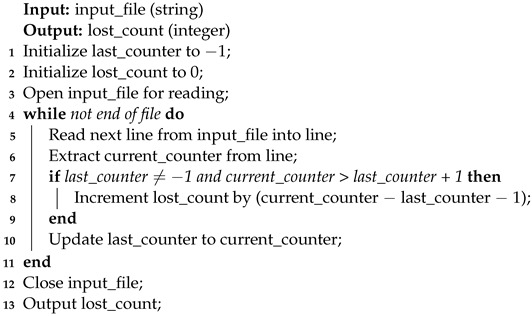



**Algorithm 8:** Message Counter—source code algorithm.

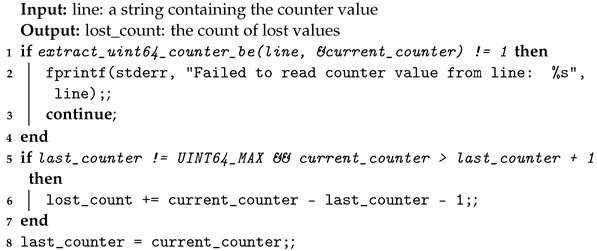



In line 10 of Algorithm 9, the resulting Argon2 hash is used to populate the 96-bit derived key with pseudo-random data obtained from the serial number of the device.

The serial number is also used to generate pseudorandom values for the key scheduling algorithm in [41] at lines 15, 20, and 25. These lines use Algorithm 4 in order to initialize the 3 substitution boxes $Sbox_{1}$, $Sbox_{2}$, and $Sbox_{3}$.

The asynchronous Speck-R encryption and decryption function developed in [53] is shown in Algorithm 10. In the counter-mode-based block cipher, the encryption and decryption are performed by the same function. The asynchronous version of Speck-R derives this counter value in left-side ($N_{L}$) and right-side ($N_{R}$) words based on the current location of the block to be encrypted or decrypted, called datasize—see line 2 of Algorithm 10. This function was designed to cope with datagram packets like UDP that do not always arrive in the same order in which they were sent.

The position indicated by the variable datasize and the values of the counters it1 and it2 involved in updating the dynamic substitution boxes must be perfectly synchronized between the sender and the receiver of the log messages; that is, both ends of the data diode have to be perfectly synchronized, meaning that the first message sent by the sender is going to be the first message received by the log concentrator. This is important also when rebooting the device.
**Algorithm 9:** Initialization of the Speck-R context (speckr_init function).
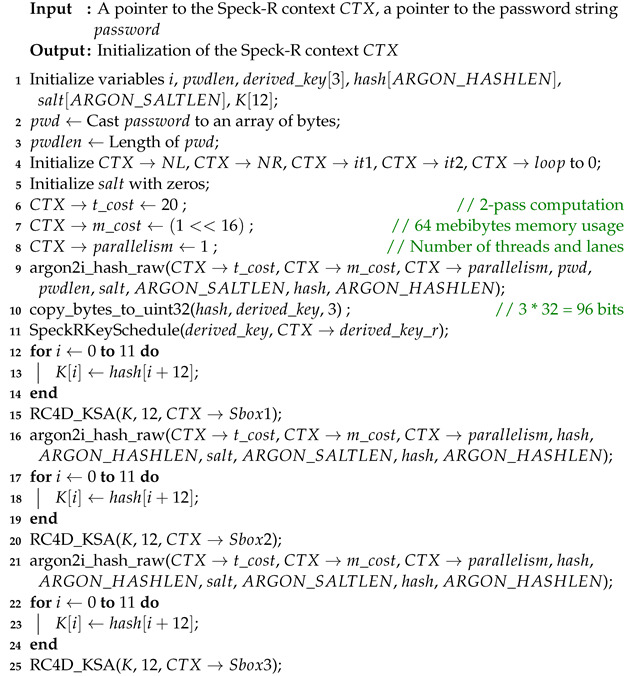


The log message counter is 64 bits to avoid overflowing. The upper-bound value for this counter is greater than $1.84\times 10^{19}$. The counter is not part of the real log message; it is only used for deduplication of packets. In line 3 of Algorithm 10, the context counters CTX−>NR and CTX−>NL are manually initialized based on the current datasize value.

Efforts have been made to ensure that the source code in [53] for the log amplifier and also in [39] for the improved implementation of Speck-R are independent of endianness; that is, no matter if the sender or the receiver is compiled on hardware with incompatible endianness, the system for amplifying and deamplifying encrypted logs through the data diode still works.

Figure 9 summarizes the flow chart for the two programs: (a) the sender on the input side of the data diode and (b) the collector on the high-security network side.

The two software parts have flowcharts that are drawn side by side to make it very visible that the decryption process depends on the synchronization of the 64-bit counter on both sides. Both programs need to start the encryption of log messages using the same counter.
**Algorithm 10:** SpeckREncrypt_async function.
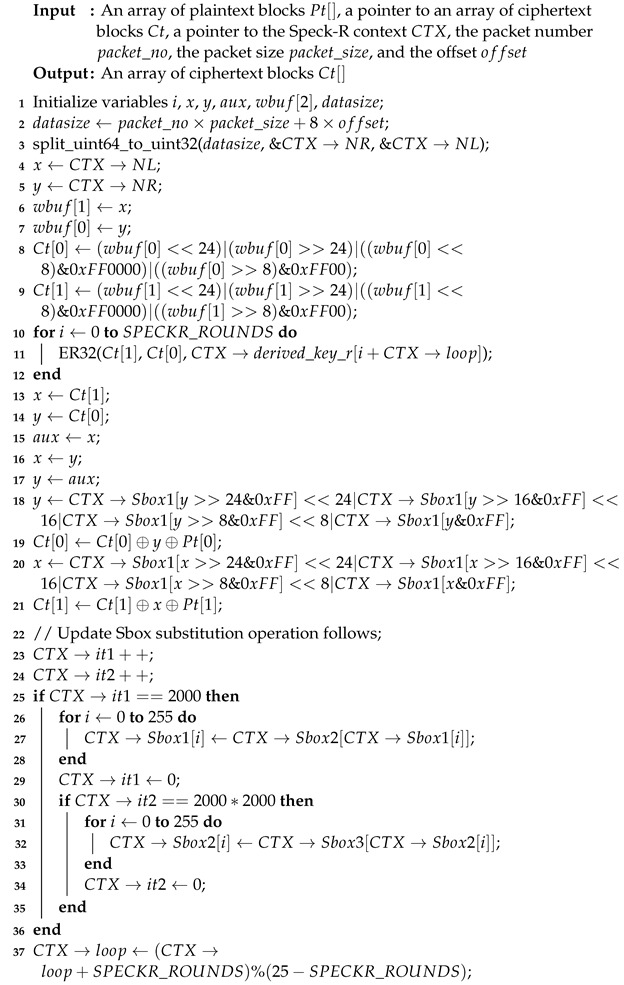


The amplifier software part listens on port 1514 on a host IP within the low-security network, and the deamplifier listens by default on port 2514 somewhere within the high-security network.

In order to enable the data diode syslog amplifier to receive messages, a client syslog concentrator from the low-security network should be configured to send data into it:


In /etc/rsyslog.conf:



. @datadiode-amplifier-host-ip:1514


On the other end of the data diode, inside the higher-security network, a local rsyslog UDP server should listen for the deamplified and decrypted messages:


In /etc/rsyslog.conf:



module(load="imudp") input(type="imudp" port="514" Address="127.0.0.1")


If this server is not the local host, the “Address” parameter must take an IP from within the internal network to make itself reachable by the receiver software part. Similar settings must be added for other flavors of the syslog service.

The next section presents the resulting measurements obtained using the software and the setups previously explained.

## 3. Results

### 3.1. Packet Loss Mitigation

In order to measure the packet loss and the efficiency of the repeatfactor over the unidirectional link, two laptops with Intel i7 running rsyslog [51] and the Github software [53] explained in the previous section were connected using the two available data diodes: the FastEthernet (FE) data diode and the GigabitEthernet (GbE) data diode. Table 1 presents the message loss observed using these settings when the receiver is flooded with packets for various values of repeatfactor.

It is evident that, when the receiver is flooded, increasing the value of the repeatfactor reduces the number of lost packets by more than half.

In regular use cases where packets are sent at 100 ms intervals with a speed of 73.5 Kbps, 0 messages are lost. Also, at 1 ms intervals, the speed increases to 6.8 Mbps and no UDP packet is actually lost. Packet loss only appears at higher rates when the receiver is flooded with messages and it is shown that the retransmission of packets successfully mitigates for the packet loss over the data diode. This is only necessary in congested networks.

Data loss is not something specific to unidirectional links. All UDP transmissions can be expected to lose some of the packets. Well-behaved programs are expected not to flood the syslog daemon, and regular use cases fit into this category.

### 3.2. Randomness Tests for Speck-R Enhanced

The randomness tests in Table 2 were performed to evaluate the randomness of 1024 bitstreams of 1,048,576 bits that resulted in the encryption process of a one gigabyte zero-byte file made from /dev/zero. The zeroed file is encrypted using the “encrypt.c” tool from [48] in counter mode. The result should be indistinguishable from random data obtained from a true random number generator (TRNG) or pseudorandom data obtained from /dev/urandom. The file with zeroed values is produced by reading /dev/zero with a tool called “dd”:


dd if=/dev/zero of=zero.bin bs=1M count=1024


The NIST statistical test suite (STS) has been used to validate the behavior of reduced-round ciphers when they are expected to produce pseudorandom data [57]. The implementation used is an improved version of the original source code with version 2.1.2 [58].

The randomness tests in Table 3 reflect how many of the 11 AIS31 tests [59,60] implemented on GitHub by [61] did not pass as random data having a green column associated by the tool.

It is clearly visible from both Table 2 and Table 3 that the Speck-R implementation is superior to SPECK6496 in terms of producing random data in reduced rounds. In other words, both statistical test suites show that the Speck-R cipher converges faster toward what random data look like when the number of encryption rounds is increased.

The NIST STS specifies that it is normal to expect some of the tests to fail from time to time, even on random data. This appears in agreement with the number of 0–3 failed tests out of 188 in Table 2. However, the five files produced using the OneRNG USB stick [56] truly show what random data look like, having 0 failed tests.

## 4. Discussion

### 4.1. Packet Loss Mitigation

Table 1 from the previous section indicates that, when the receiver is flooded with packets, causing serious packet loss due to the nature of non-guaranteed delivery over UDP data-grams, 90.71% of the 30 bytes messages are dropped. For the gigabit link, 90.89% of the 1 Kb messages are lost.

When repeatfactor increases to 99, which means that each message is repeated 99 times, 43.43% of the messages are lost on the FastEthernet link, and 43.76% of the packets are lost on the gigabit link.

This is evidence of the fact that packet loss mitigation works, although the best practice is always not to flood the link with packets. For this reason, if packet loss is observed even when repeatfactor takes values larger than 50, it is recommended that the unidirectional link be scaled by adding a second and a third data diode to cope with the traffic.

### 4.2. Randomness Tests for Speck-R Enhanced

Table 2 and Table 3 show that, after just four rounds of encryption, our enhanced Speck-R implementation [48] produces ciphertext that cannot be differentiated from random data. SPECK [47] only looks random after seven encryption rounds. This indicates that Speck-R manages to achieve the same level of security with fewer iterations.

The reference OneRNG [56] true random data pass all statistical tests with flying colors. The AIS31 [59,60] and NIST STS [57] standards are widely used to validate cipher output. The purpose of using ultra-lightweight cryptography is to enable encrypted syslog messages for resource-constrained information producers.

### 4.3. Maintenance Issues

A media converter is an active network device that is used to bridge two different types of media, such as copper ethernet and fiber ethernet. Although copper wiring is suitable for short-range connections, fiber optic cables are the preferred choice for long-distance communication. Media converters are usually used as network extenders.

Media converters may be managed or unmanaged. The latter case does not provide the same level of monitoring, fault detection, and configuration as managed media converters.

A managed media converter is equipped with a remote web interface or simple network management protocol (SNMP) and is the recommended choice for datacenters and large networks. Managed media converters have built-in test modes and fiber-fault alerts. The SMI-10G-STS-managed media converters can update their firmware via PerleVIEW, a compatible management system that can also post diagnostic information directly on social networks [62].

Paired unidirectional media converters hardened for extreme industrial temperatures such as M/GE-PSW-SFP-01-URX are purely data diodes and are not commodity products [63].

Media converters with a disconnected receive (RX) port, as demonstrated with the MC100CM model [64] and with others [65], will automatically turn off unless electronic hacks are in place. In this case, the solution is not an off-the-shelf approach. We believe that everyone should be able to create affordable hardware data diodes.

Network monitoring and diagnosis are very important in high-security networks. However, a data diode with management functions is also a security risk. The highest security data diode enforces unidirectionality based on physical properties and hardware and is completely immune from software exploits.

The media converters used in this paper are unmanaged. For resource-constrained devices, a managed media converter simply adds another layer of complexity and vulnerability.

Small-form-factor pluggable optical transceivers (SFPs) can be easily interchanged between fiber optic media converters. SFP transceivers provide maximum flexibility as they can be replaced or updated as needed.

The mean time before failure (MTBF) of a media converter is comparable with that of a network card. Data diodes made of multiple media converters clearly introduce multiple points of failure.

The only way to diagnose the failure of a data diode made of unmanaged media converters is to manually check the devices and cables. This is very uncomfortable when the network is equipped with advanced diagnostic and monitoring tools.

The latency introduced by media converters is negligible as their primary function is to convert signals from one medium to another. Delays of the order of nanoseconds are simply not noticeable.

When using long-range media converters while the data diode requires a connection over a short fiber optic cable, optical attenuators should be deployed in order to reduce the strength of the light signal.

Media converters work on low power and it is desirable that they are protected by UPS systems. For example, the MC200CM media converter uses DC 9 V 600 mA with standard center-positive barrel connectors. The default power supply provided by the vendor has a real output range of 9.1–9.2 V but it is not a high-performance power source. A stabilized power source for 9V is an order of magnitude more expensive than the media converter itself. For example, the stabilized XP POWER ECE60US09-S 9V power source has an output of 6.67 A and can be used to power multiple data diodes at once [66]. For example, Figure 10b shows two XP POWER ECE60US09-Ss powering up six data diodes that do not require active cooling in a 2U rackable case. Several data diodes can be used to scale multiple unidirectional gigabit links between networks using multiple network ports or IP aliases on both sides. The listener program binds the same port on any IP, so incoming packets are spread over several links using a round-robin algorithm in order to scale the entire system as in Figure 10a.

### 4.4. Insider Threats

If an attacker is able to breach the high-security network, write-once read-many (WORM) storage will protect the already stored log messages but the live tailing and the monitoring are compromised since the attacker can feed the log monitor SIEM or SOAR with whatever they want. This is why the use case of a data diode for a unidirectional link between the log inspection network and the log creator network is very useful. An attacker simply cannot breach a network that lies behind a data diode since there is no way to communicate back and the software tools provided by [53] only decrypt, deduplicate, and relay data to a syslog server [51] internal to the high-security network.

An insider is a person who has authorized access inside a higher-security network: an employee with necessary clearance.

Insider threats may arise from two types of actions: unintentional and intentional. Unintentional acts can be further classified as negligence or accidents [67]. Potential threats and mitigation techniques are discussed.

#### 4.4.1. Unintentional

Unintentional threats consist of the non-malicious exposure of infrastructure and data [67]. For example, a system administrator with physical access to the data diodes can rewire the cables, creating a bidirectional connection. The management interface of a media converter may be mistakenly connected to the wrong VLAN that provides access to unauthorized users.

These threats can be discovered and mitigated by following standard procedures and performing regular audits and security evaluations. Managed media converters and network devices can be used to detect and remove unintentional threats.

#### 4.4.2. Intentional

From the list of intentional threats, the most dangerous is the collusive one [67]. Based on the idea that an insider collaborates with external adversaries, several techniques can be used to create covert channels and exfiltrate the data. Malware has been demonstrated to manipulate network card light-emitting diodes (LEDs) [68], ethernet cables [69], or powerlines [70]. Mobile phones near air-gapped computers can be used to exfiltrate data using malware running specific memory instructions on the infected computer [71].

In order to prevent such threats, the CISA recommends an internal mitigation program for organizations [67]. On the technical side, measures can be taken, such as installing software that detects when there is any abnormal change in screen brightness [72] or when air-gapped environments start to present suspicious activity, which can be detected with artificial-intelligence-enhanced sensors.

Running antivirus software on air-gap systems is problematic because software and signature updates will have to bridge the air gap and the Internet.

### 4.5. Logging Best Practice

The NIST SP800-92r1 [73] standard outlines several key considerations for effectively managing log sources. For example, organizations should first classify each log source by determining whether logging is required, recommended, not recommended, or prohibited. Next, they need to identify the types of events that should or must be logged by each source, as well as the events that should not or must not be logged.

A vital step is assessing whether log sources might inadvertently capture sensitive data, such as personal information, passwords, or access tokens. The frequency of event logging must also be established, along with guidelines for handling log generation errors. Proper clock synchronization across log sources is necessary in order to provide a unitary perspective on time.

Adequate log storage space must be ensured in both log sources and infrastructure, and plans must be made to handle storage and transfer errors. Internet of Things devices do not have the capacity to store logs for a long period of time and are prone to firmware update bugs and misconfiguration.

## 5. Conclusions

The implementation of a secure log transmission system using a data diode was presented, addressing the challenges associated with transmitting syslog messages over standard network protocols like UDP, which lack reliability and security. The proposed solution involves an encrypted syslog redirector and amplifier designed to enhance the security and reliability of log data transmission, particularly in environments where sensitive information is at risk of interception.

The main contribution consists of robust free software made available on Github [53] together with the only public implementation of the Speck-R [39] cipher, also released on Github with several fixes and improvements [48].

In terms of reducing packet loss, the results show that 90% of the packets are lost during network congestion. The mitigation technique requires that log messages are repeated. When the coefficient repeatfactor is higher, the number of lost packets is reduced by more than 50%. The results are similar for both FastEthernet and gigabit data diodes.

Ultra-lightweight cryptography implemented for log security over the wire is shown to produce random-looking output more efficiently compared to NSA’s SPECK cipher [47]. Results have shown that, after only four rounds, our improved version of Speck-R passes the German (European) AIS31 randomness standard tests but also NIST’s statistical test suite (STS).

The construction of reliable data diodes using commodity off-the-shelf hardware (affordable media converters and fiber optic splitters at a fraction of the price of patented closed-source data diode products) simply makes this proof of concept a reliable solution for the network segregation of logs, which is suitable for any environment where logs are critical.

The results have shown that, in regular use cases, where most of the bandwidth of the data diode is free, amplification of the log messages is not mandatory, which means that repeatfactor can be set to 1. On the other hand, when the data diode throughput is congested, because the sender is flooding the receiver with many packets per second, an increase in repeatfactor successfully mitigates packet loss.

A data diode ensures that log data cannot flow in the wrong direction, preventing any reverse communication and significantly enhancing security against unauthorized access. By isolating the log network from potentially insecure environments, a data diode protects sensitive log data from interception and tampering. The use of our data diode software helps to maintain the integrity and availability of log data, reducing the risk of data loss and ensuring reliable log transmission even over insecure networks.

This research highlights the importance of secure log management in modern digital environments and proposes a cost-effective solution that relies on free software and reproducible results. The implementation of a data diode, combined with robust encryption methods, offers a reliable means of protecting sensitive log data against potential threats.

Ciphers where the round function relies on exclusive-or, modular addition, and rotation operations are called ARX (addition–rotation–xor) ciphers. They are meant to be fast and easy to implement in the Internet of Things and resource-constrained devices. The original SPECK [47] cipher was designed to be implemented in software libraries, and Speck-R [39] is a reduced-round version that also involves RC4 [41] and confusion–diffusion substitution boxes. These boxes require tables to be stored and dynamically interchanged in memory, which means that the implementation is a hybrid ARX cipher [48]. The analysis performed using NIST [57] and AIS31 [60] emphasizes that the reduced-round Speck-R is superior to the original SPECK6496, which is ARX.

### Future Work

The amplification factor aims to prevent message losses over the data diode. Since there is no feedback loop, messages can be dropped for various reasons. Extensive tests have shown that, on non-real-time networks, so unlike isochronous real-time (IRT) communication, the receiver simply may miss a packet delivered by the sender because there is no guarantee for the delivery and no synchronization in terms of sending and receiving packets using some sort of flow control algorithm to prevent data loss or a form of synchronization based on precise timing.

Although this is true, the amplification factor only diminishes packet loss when the data diode is congested, i.e., when the sender is flooding the receiver with log messages. Depending on the number of packets per second (PPS) that the sender generates when repeating log messages, repeatfactor can be diminished to very small values. In the usual case, when most of the link bandwidth is free, the amplification of the log messages is simply optional.

However, when artificial delays between log messages cannot be tolerated by the large amount of incoming packets per second in the sender network, repeatfactor scales up to improve the reception of the packets on the de-amplifier side.

An alternative would be to create a queue of the messages partially stored in a local file and delay the reception of the syslog information as long as it is necessary to avoid flooding the receiver.

This brings to mind that, on a hostile low-security network, the amplifier software should develop defensive mechanisms in order to mitigate for internal abuses like UDP floods with forged values for the message counter.

The ultimate goal of this work is to provide an affordable, off-the-shelf solution for concentrating logs through a hardware data diode in a secure fashion, relying solely on free software and reproducible results. Otherwise, expensive data diode software and hardware may provide patented techniques and technologies that achieve the same level of physical security.

Encrypting logs over virtual private networks (VPNs) is a good practice, but VPNs are bidirectional, so this simply does not work over hardware data diodes.

Argon2 [54] is the winner of the Password Hashing Competition [55] and is a very robust choice for increasing password security against brute force and creating pseudorandom outputs from simple strings. An alternative for devices with very limited resources is BLAKE3 [74,75,76], having seven rounds, just like the Speck-R algorithm [39].

In order to use the Speck-R implementation from [48] in low-level communication protocols like near-field communication (NFC), it has to be integrated with basic encoding rule libraries like BERLib [77]. 

## Figures and Tables

**Figure 1 sensors-24-06537-f001:**
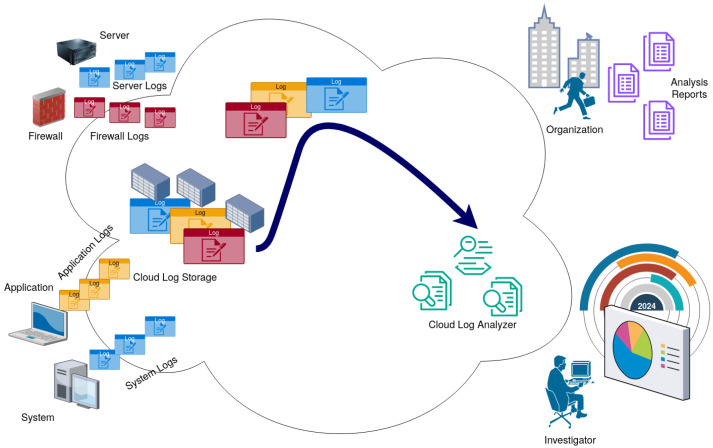
Generalized cloud log forensics as proposed in [21] (reconstruction of Figure 4 Page 12).

**Figure 2 sensors-24-06537-f002:**
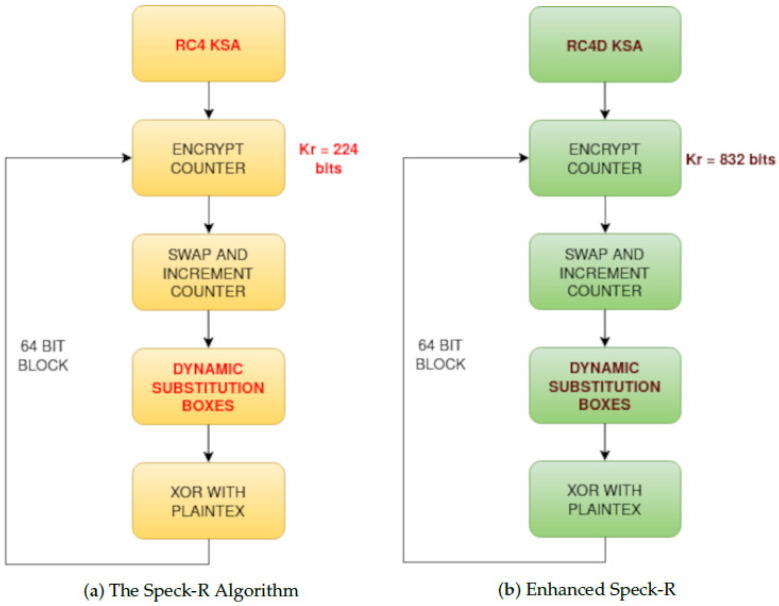
Differences between Speck-R [39] (**a**) and Enhanced Speck-R (**b**).

**Figure 3 sensors-24-06537-f003:**
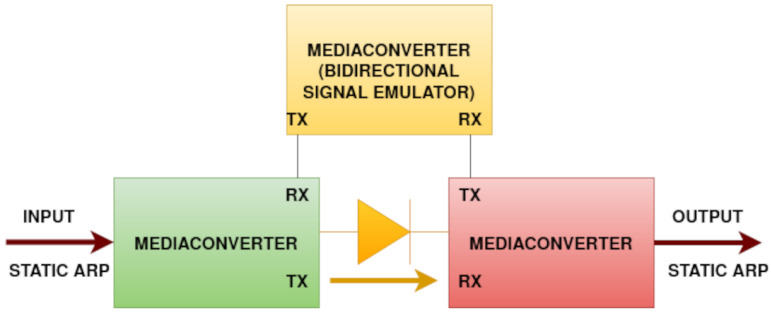
Connection schematics for a gigabit ethernet data diode using 3 off-the-shelf media converters.

**Figure 4 sensors-24-06537-f004:**
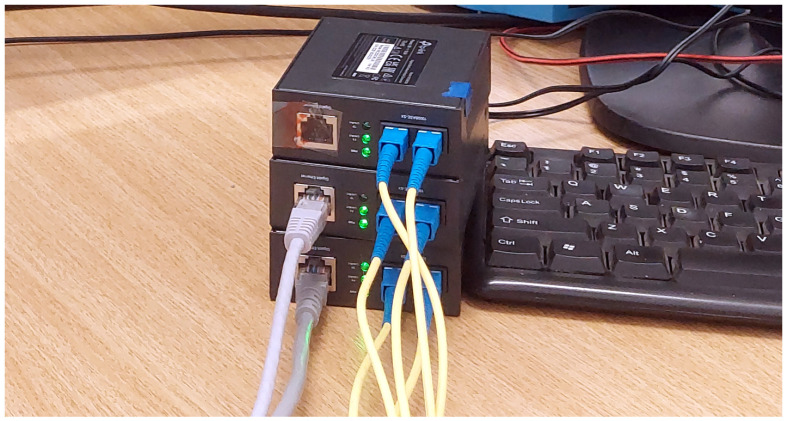
Gigabit ethernet data diode using 3 off-the-shelf media converters.

**Figure 5 sensors-24-06537-f005:**
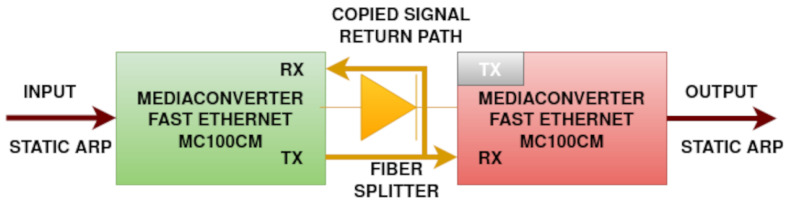
Fast ethernet data diode schematics made using only 2 media converters.

**Figure 6 sensors-24-06537-f006:**
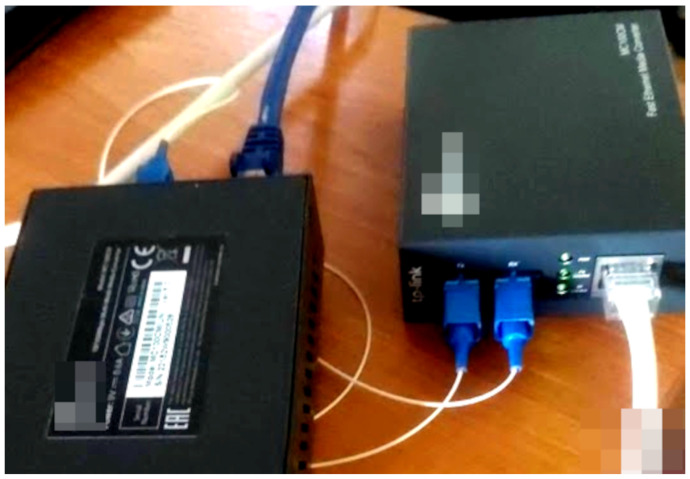
Fast ethernet data diode with 2 media converters.

**Figure 7 sensors-24-06537-f007:**
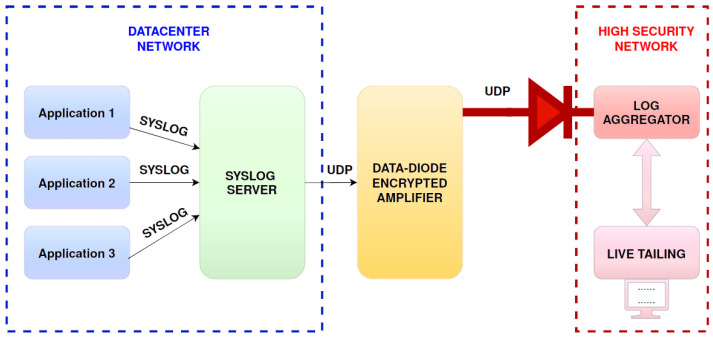
Solution architecture for safely concentrating logs via UDP through a data diode for live tailing with an SIEM or SOAR in a network with a higher security level.

**Figure 8 sensors-24-06537-f008:**
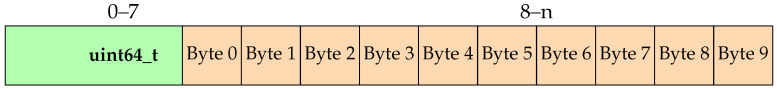
Line message format where the first 8 bytes are a uint64_t counter and the rest are a list of printable characters.

**Figure 9 sensors-24-06537-f009:**
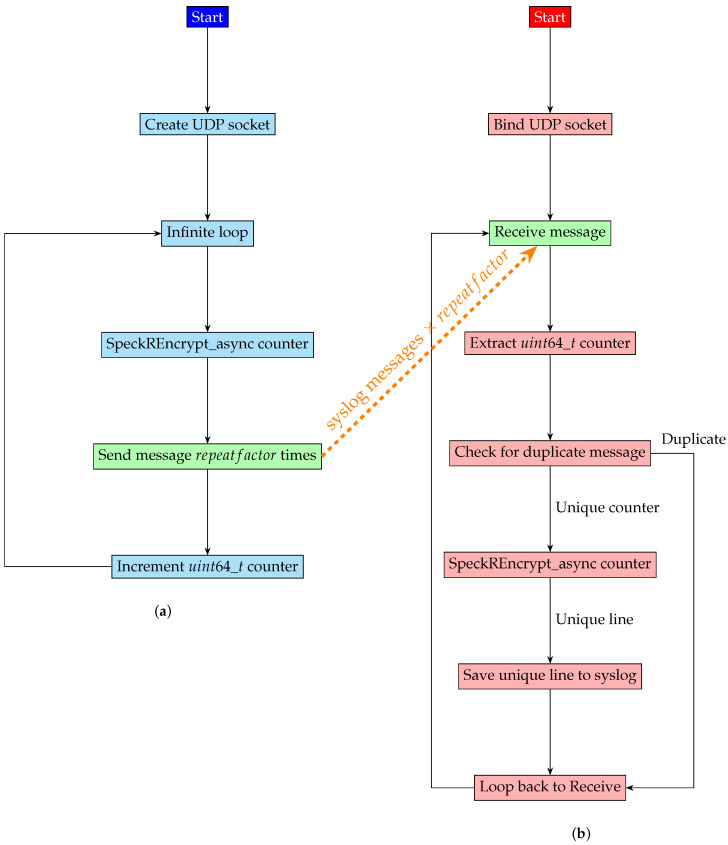
Sender and receiver programs. (**a**) Sender program in Algorithm 5; (**b**) receiver program in Algorithm 6.

**Figure 10 sensors-24-06537-f010:**
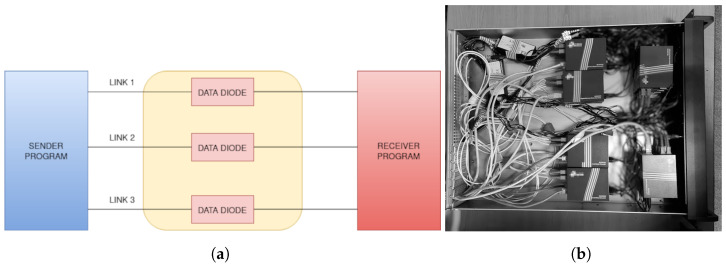
Combined data diodes for scaling the program over several unidirectional links. (**a**) parallel unidirectional connections used in a round-robin fashion; (**b**) several self-made gigabit data diodes in a 2U rack case.

**Table 1 sensors-24-06537-t001:** Syslog message loss through the unidirectional link when the receiver is flooded.

repeat factor	0	4	49	99	Speed	Message Size
FastEthernet	95,120,439	94,084,149	70,937,632	45,544,804	71 Mbps	30 bytes
Gigabit	9,530,579	9,412,165	7,114,333	4,588,976	930 Mbps	1 Kb
Total Number of messages: 104,857,600 (FE) and 10,485,760 (GbE)						

**Table 2 sensors-24-06537-t002:** Failed randomness tests from the NIST Statistical Test Suite for reduced-round SPECK6496 versus the enhanced Speck-R implementation.

	Rounds														
	1	2	3	4	5	6	7	8	9	10	11	12	13	14	15
SPECK	188	187	187	23	92	7	0	0	1	2	0	3	1	0	1
OneRNG TRNG	0	0	0	0	0	0	0	0	0	0	0	0	0	0	0
Speck-R	27	30	81	1	2	0	0	2	2	2	1	0	1	0	0

The NIST statistical test suite version 3.2.6 has 188 randomness tests. The OneRNG TRNG [56] was used to produce five files, each of 1 Gb size, passing all randomness tests, and is used as a reference.

**Table 3 sensors-24-06537-t003:** Failed randomness tests from the AIS31 Statistical Test Suite for reduced-round SPECK6496 versus the enhanced Speck-R implementation.

	Rounds														
	1	2	3	4	5	6	7	8	9	10	11	12	13	14	15
SPECK	9	9	6	3	5	1	0	0	0	0	0	0	0	0	0
OneRNG TRNG	0	0	0	0	0	0	0	0	0	0	0	0	0	0	0
Speck-R	7	5	3	0	0	0	0	0	0	0	0	0	0	0	0

The AIS31 statistical test suite has 11 global tests, some containing many sub-tests. The OneRNG TRNG [56] was used and is used again as a reference for truly random data.

## Data Availability

The data associated with this article are fully reproducible. The true hardware number generator data and the analysis provided by statistical tests for encrypted files are available at https://staff.cs.upt.ro/~alin.anton/BANPUMP/datasets/syslog accessed on 2 June 2024. Data are available under the terms of the Creative Commons Zero “No rights reserved” data waiver (CC0 1.0 Public domain dedication).

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
