# Peer review of "Enhancing Syslog Message Security and Reliability over Unidirectional Fiber Optics"

_sensors, 2024, doi:10.3390/s24206537_

Round 1

Reviewer 1 Report

Comments and Suggestions for Authors

The topic presented in this paper is not well aligned with the scope of the Sensors Journal. The contribution of this paper is in the software layer on a well-known approach for secure hardware channels called data diodes. I recommend reconsidering the submission to other journals such as the Journal of Cybersecurity and Privacy.

Comments on the Quality of English Language

Author Response

Comments and Suggestions for Authors

R1: The topic presented in this paper is not well aligned with the scope of the Sensors Journal. The contribution of this paper is in the software layer on a well-known approach for secure hardware channels called data diodes. I recommend reconsidering the submission to other journals such as the Journal of Cybersecurity and Privacy.

ANSWER:

Dear Reviewer,

Thank you for the time taken to review our work and for the suggested journal.

We did a major revision of English and structure. Several subsections have been added to make the text more comprehensible and more useful to the readers.

The first paragraph in the Section 1.1 “Related Work” has been added to better emphasize related articles from the Sensors Journal: [7] – air-gaps and data diodes, [8-9] lightweight cryptography, [11] block-cipher counter-mode encryption, which we also use, and others. We sincerely believe this paper is suitable for the Sensors journal.

The contribution of our paper is not only in the software layer, we believe that everyone should be able to benefit from affordable hardware data diodes, and we added references [63] and [64] explaining why the connection schematics that they use do not work without hacking the media-converter electronics. The optical receiver port on modern equipment will shut down the link if there is no light detected at the proper strength and frequency. Our goal is to empower the user to create data diodes with off-the-shelf media-converters, with the practical application of protecting logs. Resource-constrained devices do not have the capacity to aggregate logs from multiple sources and analyze them properly.

Respectfully,

The Authors

Reviewer 2 Report

Comments and Suggestions for Authors

The manuscript “Enhancing Syslog Message Security and Reliability Over Unidirectional Fiber Optics” mainly studies the secure log transmission system, and proposes a novel approach to enhancing the security and reliability of syslog message transmission using unidirectional fiber optics. The proposal to use affordable components to create a data diode is innovative and addresses a significant gap in the current syslog transmission methods, which often lack robust security measures. The research is well-conducted, with a clear methodology that includes the design and implementation of an encrypted syslog redirector and amplifier. The use of encryption protocols and the mitigation of replay attacks through synchronization are sound strategies. The authors have also provided software and datasets for reproducibility, which is a strong point in the scientific method.

The research topic is interesting and within the scope of the journal. However, the authors should address the following comments in their manuscript to increase the quality of the paper.

1) Security Analysis: While the paper addresses the security enhancements, a deeper analysis of potential vulnerabilities and how they are mitigated could strengthen the manuscript.

2) Comparison with Existing Work: It is recommended that the authors provide a more detailed comparison with current syslog security solutions to better position their contribution within the existing body of research. This will help readers understand the novelty and practical implications of the research.

3) Performance Evaluation: The performance evaluation is limited to packet loss and randomness tests. It would be beneficial to include a broader range of performance metrics, such as latency and scalability.

4) Real-world Application: The authors might consider discussing potential real-world deployment scenarios and any challenges that might be encountered in such environments.

Author Response

Comments and Suggestions for Authors

The manuscript “Enhancing Syslog Message Security and Reliability Over Unidirectional Fiber Optics” mainly studies the secure log transmission system, and proposes a novel approach to enhancing the security and reliability of syslog message transmission using unidirectional fiber optics. The proposal to use affordable components to create a data diode is innovative and addresses a significant gap in the current syslog transmission methods, which often lack robust security measures. The research is well-conducted, with a clear methodology that includes the design and implementation of an encrypted syslog redirector and amplifier. The use of encryption protocols and the mitigation of replay attacks through synchronization are sound strategies. The authors have also provided software and datasets for reproducibility, which is a strong point in the scientific method.

The research topic is interesting and within the scope of the journal. However, the authors should address the following comments in their manuscript to increase the quality of the paper.

Request 1) Security Analysis: While the paper addresses the security enhancements, a deeper analysis of potential vulnerabilities and how they are mitigated could strengthen the manuscript.

Answer 1) Subsection 4.4 “Insider Threats” within the “Discussion” section has been added to provide a deeper analysis of the possible threats and mitigation techniques for the proposed solution.

Request 2) Comparison with Existing Work: It is recommended that the authors provide a more detailed comparison with current syslog security solutions to better position their contribution within the existing body of research. This will help readers understand the novelty and practical implications of the research.

Answer 2) The Introduction was extended with two subsections, “1.1 Related Work” and “1.2 Standards” in order to carefully present the broader context of our contribution within the existing body of research. Several other log security solutions have been introduced.

Request 3) Performance Evaluation: The performance evaluation is limited to packet loss and randomness tests. It would be beneficial to include a broader range of performance metrics, such as latency and scalability.

Answer 3) Unidirectional communication implies that packets do not travel in the wrong direction, which means, measuring the latency of a unidirectional link is not straightforward. Packets will not return in order to measure the response time. Media-converters are active network devices with negligible impact (nanoseconds) on packet delivery due to conversion from one medium to another (copper to fiber optics and vice-versa). This was properly discussed in the paper, and we are grateful for your suggestion.

The scalability is also discussed now, and Figure 10 was introduced to better explain how to scale the program.

Request 4) Real-world Application: The authors might consider discussing potential real-world deployment scenarios and any challenges that might be encountered in such environments.

Answer 4) Figure 10b was inserted to show 6 online data diodes grouped into a 2U rack case with a stabilized power source. This can be used to send encrypted log messages through parallel data diodes connected to the same collector. This was explained in the paper. Deployment scenarios and challenges are also discussed in the Discussion section.

We are grateful for all the suggestions provided and we think that, by implementing the modifications requested, a much better paper is now submitted.

Respectfully,

The Authors

Reviewer 3 Report

Comments and Suggestions for Authors

In this manuscript, there are some significance or novelty as follows.

‧            The proposed use of Data Diode to enhance the security and reliability of Syslog message transmission is an innovative point among the existing log transmission methods. The unidirectional data flow characteristic of Data Diode effectively prevents unauthorized access and eavesdropping, and ensures the integrity and confidentiality of sensitive log data.

‧            A lightweight encryption algorithm (SPECK-R) was adopted and improved to make it suitable for resource-constrained devices. This improvement not only improves the efficiency of the encryption algorithm, but also ensures that outputs with high randomness can be produced despite the reduced number of encryption rounds.

‧            Free software and datasets are provided to facilitate the reproduction of experimental results by other researchers and practitioners. This openness and reproducibility is important for promoting academic research and practical applications.

The other results will be some potential reasons for acceptance, such as

‧            The research addresses security and reliability issues in Syslog message transmission, especially message loss and unencrypted transmission when using the UDP protocol. By introducing data diodes and encryption algorithms, the security of log transmission is significantly improved.

‧            Proposed solution is highly practical and operable, using inexpensive off-the-shelf components (e.g., media converters and fiber optic splitters), which makes the solution cost-effective in practical applications.

‧            Study provides detailed experimental results and performance evaluations that demonstrate the effectiveness of the proposed scheme in terms of reliability, security and throughput. These data provide strong support for the feasibility of the scheme.

However, there are some reasons for rejection on this manuscript, mainly in the form of

1. The improvements and implementation details of the encryption algorithms described in the paper are more complex and may be difficult for some readers to understand and reproduce.

2. Although the paper proposes the use of data diodes, the configuration and maintenance of data diodes in real-world applications may introduce additional complexity and cost, which is not fully discussed in the paper.

3. the discussion of potential attacks and the description of defense measures in the paper are rather brief, especially for the lack of in-depth analysis and response strategies for internal attacks and misuse behaviors in high-security networks.

Those above must be analyzed clearly. To enhanced the quality of the paper, therefore, this reviewer proposed improvement as follows.

4. Add detailed explanations and sample code for the improved parts of the encryption algorithm to help readers better understand and implement these improvements. Consider providing more diagrams and flowcharts to illustrate the encryption and decryption processes.

5. Add a detailed discussion of data diode configuration and maintenance to the paper, including problems and solutions that may be encountered to aid deployment and management in real-world applications.

6. Expand the discussion of potential attacks and defenses, especially for internal attacks and abuses in high-security networks, to provide more detailed analysis and response strategies. This will help improve the comprehensiveness and practicality of the program.

Author Response

However, there are some reasons for rejection on this manuscript, mainly in the form of

1. The improvements and implementation details of the encryption algorithms described in the paper are more complex and may be difficult for some readers to understand and reproduce.

Algorithms 1 and 2 and Algorithms 3 and 4 have been introduced for side-by-side comparison to explain in detail why our Enhanced version is superior and why the original version is broken. The enhanced version of Speck-R uses 208 bits for the round key instead of the truncated 56 of the original cipher. Also the substitution boxes are generated using a secure version of RC4 – RC4D. The cipher modifications are emphasized using blue and red colors in Algorithms 1–4 and Figure 2 compares the two new flocharts side by side.

2. Although the paper proposes the use of data diodes, the configuration and maintenance of data diodes in real-world applications may introduce additional complexity and cost, which is not fully discussed in the paper.

Subsection 2.2 “Data Diode Configuration” and Subsection 4.3 “Maintenance Issues” have been inserted in order to fully discuss real-world situations and possible solutions to the problems that one will face trying to deploy our solution.

3. the discussion of potential attacks and the description of defense measures in the paper are rather brief, especially for the lack of in-depth analysis and response strategies for internal attacks and misuse behaviors in high-security networks.

Subsection 4.4 “Insider Threats” was inserted discussing unintentional and intentional insider threats and possible mitigations, according to international standards but also according to technical measures for mitigation.

Those above must be analyzed clearly. To enhanced the quality of the paper, therefore, this reviewer proposed improvement as follows.

Request 1 Add detailed explanations and sample code for the improved parts of the encryption algorithm to help readers better understand and implement these improvements. Consider providing more diagrams and flowcharts to illustrate the encryption and decryption processes.

Answer 1 Algorithms 1 and 2 and Algorithms 3 and 4 have been introduced for side-by-side comparison to explain in detail why our Enhanced version is superior and why the original version is broken.

Request 2 Add a detailed discussion of data diode configuration and maintenance to the paper, including problems and solutions that may be encountered to aid deployment and management in real-world applications.

Answer 2 The Methodology section now has a “Data Diode Configuration” subsection where all the necessary steps for creating a data diode using off-the-shelf media-converters and configuring the upstream and downstream hosts are explained, as requested. Thank your for suggesting such a useful modification. The Discussion section includes Fig.10b showing real-world data diodes made of 18 off-the-shelf media-converters and explanations on how to use them for scaling the program. Problems are well discussed as requested.

Request 3 Expand the discussion of potential attacks and defenses, especially for internal attacks and abuses in high-security networks, to provide more detailed analysis and response strategies. This will help improve the comprehensiveness and practicality of the program.

The Discussion section was extended to include the requested details: in the Maintenance issues subsection the challenges and solutions are discussed; the Insider Threats subsection has a deeper analysis on threats and mitigations especially relevant for high-security networks, as requested;

We are grateful for all the suggestions provided and we think that, by implementing the modifications requested, a much better paper is now submitted.

Respectfully,

The Authors

Reviewer 4 Report

Comments and Suggestions for Authors

The authors’ implementation of a secure log transmission system using a data diode. They address the challenges associated with transmitting syslog messages over standard network protocols such as UDP, which lack reliability and security. The research highlights the importance of secure log management in modern digital environments and proposes a cost-effective solution that relies on open-source software and reproducible results.

This work is worthy of publication, but first, it needs some corrections.

-There are many acronyms without definitions in the text,

-Table captions must end with a " . ",

-Figure captions must end with a " . ",

-I believe that for a good reading of the text the algorithms should be in the appendix.

-The results and discussions section is confusing to me.

Comments on the Quality of English Language

There are many typos in this paper. Here I show these errors only for the abstract and conclusion. The authors should do a major revision of the English.

In the abstract

components is proposed-->components are proposed
protocol which requires that-->protocol that requires that
synchronized mitigating for replay attacks-->synchronized, mitigating replay attacks
can mitigate for UDP packet loss but this -->can mitigate UDP packet loss, but this
random looking output even on reduced number-->random-looking output even on a reduced number
available for reproducing the results-->available to reproduce the results
.
In the conclusion

off the shelf-->off-the-shelf
 mediaconverters--> media-converters
segregation of logs suitable for any-->segregation of logs, which is suitable for any
shown that on regular-->shown that in regular
is congested, because-->is congested because
packet loss but more -->packet loss, but more
modular addition and rotation-->modular addition, and rotation
internet of things-->Internet of Things
memory which means-->memory, which means
 clearly have shown--> clearly has shown
SPECK6496 which is a-->SPECK6496, which is a

Author Response

Comments and Suggestions for Authors

The authors’ implementation of a secure log transmission system using a data diode. They address the challenges associated with transmitting syslog messages over standard network protocols such as UDP, which lack reliability and security. The research highlights the importance of secure log management in modern digital environments and proposes a cost-effective solution that relies on open-source software and reproducible results.

This work is worthy of publication, but first, it needs some corrections.

-There are many acronyms without definitions in the text,

-Table captions must end with a " . ",

-Figure captions must end with a " . ",

-I believe that for a good reading of the text the algorithms should be in the appendix.

-The results and discussions section is confusing to me.

Answer 1: The paper has been modified as requested. 

Comments on the Quality of English Language

There are many typos in this paper. Here I show these errors only for the abstract and conclusion. The authors should do a major revision of the English.

Answer 2: The paper was extensively revised for English and typos.

In the abstract

Answer 3: The paper was modified as requested.

components is proposed-->components are proposed
protocol which requires that-->protocol that requires that
synchronized mitigating for replay attacks-->synchronized, mitigating replay attacks
can mitigate for UDP packet loss but this -->can mitigate UDP packet loss, but this
random looking output even on reduced number-->random-looking output even on a reduced number
available for reproducing the results-->available to reproduce the results
.
In the conclusion

Answer 4: The paper was modified as suggested.

off the shelf-->off-the-shelf
 mediaconverters--> media-converters
segregation of logs suitable for any-->segregation of logs, which is suitable for any
shown that on regular-->shown that in regular
is congested, because-->is congested because
packet loss but more -->packet loss, but more
modular addition and rotation-->modular addition, and rotation
internet of things-->Internet of Things
memory which means-->memory, which means
 clearly have shown--> clearly has shown
SPECK6496 which is a-->SPECK6496, which is a

Thank you again,

Respectfully,

The Authors

Round 2

Reviewer 1 Report

Comments and Suggestions for Authors

Thanks for the authors for the clarifications. Since there are similar topics previously accepted, it makes sense to keep the paper within the scope of this journal. 

The paper in its current form has improved on the purpose and general readability, addressing properly the comments from previous round.

Comments on the Quality of English Language

English and readability of the paper is good, minor issues could be addressed with the editorial review. 

Reviewer 3 Report

Comments and Suggestions for Authors

Comments for the manuscript---- Sensors-3202057-peer-review-v2

In the revised manuscript, more significances are emphasized again and all comments have been responded.

1.         The proposed use of data diode to enhance the security and reliability of syslog message transmission is an innovative point among the existing log transmission methods. The unidirectional data flow characteristic of data diode effectively prevents unauthorized access and eavesdropping, and ensures the integrity and confidentiality of sensitive log data.

2.         A lightweight encryption algorithm is adopted and improved to make it suitable for resource-constrained devices. This improvement not only improves the efficiency of the encryption algorithm, but also ensures that outputs with high randomness can be produced despite the reduced number of encryption rounds.

3.         Free software and datasets are provided to facilitate the reproduction of experimental results by other researchers and practitioners. This openness and reproducibility are important for promoting academic research and practical applications.

4.         The research addresses security and reliability issues in syslog message transmission, especially message loss and unencrypted transmission when using the UDP protocol. By introducing data diodes and encryption algorithms, the security of log transmission is significantly improved.

5.         Proposed solution is highly practical and operable, using inexpensive off-the-shelf components, which makes the solution cost-effective in practical applications.

6.         Study provides detailed experimental results and performance evaluations that demonstrate the effectiveness of the proposed scheme in terms of reliability, security, and throughput. These data provide strong support for the feasibility of the scheme.

Therefore, this reviewer thinks the manuscript should be accepted for publication in present form.